# GENERATIVE LINES MATCHING MODELS

## ABSTRACT

In this paper we identify the source of a singularity in the training loss of key denoising models, that causes the denoiser's predictions to collapse towards the mean of the source or target distributions. This degeneracy creates false basins of attraction, distorting the denoising trajectories and ultimately increasing the number of steps required to sample these models.

We circumvent this artifact by leveraging the deterministic ODE-based samplers, offered by certain denoising diffusion and score-matching models, which establish a well-defined change-of-variables between the source and target distributions. Given this correspondence, we propose a new probability flow model, the *Lines Matching Model* (LMM), which matches globally straight lines interpolating the two distributions. We demonstrate that the flow fields produced by the LMM exhibit notable temporal consistency, resulting in trajectories with excellent straightness scores.

Beyond its sampling efficiency, the LMM formulation allows us to enhance the fidelity of the generated samples by integrating domain-specific reconstruction and adversarial losses, and by optimizing its training for the sampling procedure used. Overall, the LMM achieves state-of-the-art FID scores with minimal NFEs on established benchmark datasets: 1.57/1.39 (NFE=1/2) on CIFAR-10, 1.47/1.17 on ImageNet 64×64, and 2.68/1.54 on AFHQ 64×64.

Finally, we provide a theoretical analysis showing that the use of optimal transport to relate the two distributions suffers from a curse of dimensionality, where the pairing set size (mini-batch) must scale exponentially with the signal dimension.

## 1 INTRODUCTION

Diffusion models are the core engine behind many recent state-of-the-art generative models across various domains, e.g., image generation (Song et al., 2021b; Ho et al., 2020; Dhariwal & Nichol, 2021; Rombach et al., 2022), text-to-image generation (Nichol et al., 2022; Ramesh et al., 2022; Saharia et al., 2024), audio synthesis (Kong et al., 2021; Kim et al., 2021; Chen et al., 2020; Popov et al., 2021), and video generation (Ho et al., 2022; Singer et al., 2023; Liu et al., 2024b)

This gain in popularity of the underlying denoising diffusion (Sohl-Dickstein et al., 2015; Ho et al., 2020) and score-matching (Song et al., 2019; Song & Ermon, 2020; 2019) models over GANs (Goodfellow et al., 2014) is often attributed to their improved distribution reproduction (Dhariwal & Nichol, 2021), and immunity to various optimization hurdles that plague GAN training (mode collapse and forgetting (Thanh-Tung & Tran, 2020)). Nevertheless, unlike the single-step sampling of GAN and VAE (Kingma & Welling, 2014) models, the noise removal process follows non-trivial probability flow trajectories, requiring fine quadrature steps and resulting in non-negligible computational effort during inference. This ranges between hundreds of sampling steps in early methods (Ho et al., 2020) and tens in more recent ones (Karras et al., 2022).

Distilling pre-trained denoising models allows reducing this Number of Function Evaluations (NFEs) during sampling. This approach can be carried out in different ways; learning the entire sampling procedure (Luhman & Luhman, 2021), or reducing its number of steps progressively (Salimans & Ho, 2022). More recently the denoising trajectories are learned either by ensuring a consistency along successive steps (Song et al., 2023), or along arbitrary segments (Kim et al., 2024). These methods offer a significant speedup over their teacher models, nevertheless, they also inherit inefficiencies inherent to the trajectories that they replicate.

As an alternative, the probability flow matching techniques in (Lipman et al., 2023; Albergo & Vanden-Eijnden, 2023b) incorporate Optimal Transport (OT) considerations in order to produce more constant flow trajectories, requiring fewer sampling steps. Additional improvement in straightness is achieved by an iterative rectification scheme in (Liu et al., 2023; 2024a), as well as by replacing the random pairing between the source and data examples with an OT pairing (Pooladian et al., 2023; Tong et al., 2024). While improving upon traditional denoising losses, the flow fields produced by these approaches still contain false attraction basins, causing the trajectories to curve.

In this paper, we show that the ambiguous pairing between latent source noise and target data samples leads to an ill-posed regression problem, compromising the performance of key denoising models, including denoising diffusion, score- and flow-matching. At low signal-to-noise ratios, this indeterminacy in the denoising loss becomes worse and causes the denoiser's predictions to collapse toward the mean of either the source or target distributions. This creates false basins of attraction that curve and distort the denoising trajectories, ultimately increasing the number of steps needed for accurate sampling.

We avoid this singularity by leveraging the fact that certain denoising diffusion (Song et al., 2021a) and score-matching (Song & Ermon, 2019; Karras et al., 2022) models give rise to *deterministic* ODE-based flows that give rise to a well-defined change-of-variable between the source and target distributions. Unlike existing approaches that distill the underlying inefficient probability flow trajectories, we only leverage the pairing induced between the distributions. Given this correspondence, we construct a new probability flow model, the *Lines Matching Model* (LMM), which matches *globally straight* lines interpolating between the distributions. As demonstrated in Figure 1, the flow fields produced by the LMM display notable temporal consistency, resulting in trajectories with excellent straightness scores.

Beyond its sampling efficiency, and unlike other flow matching formulations, the LMM's training loss allows us to further improve the fidelity of its generated samples by incorporating domain-specific reconstruction and adversarial losses, as well as optimizing its training for the sampling procedure used. Overall, the LMM achieves state-of-the-art Fréchet Inception Distance (FID) scores using a minimal NFEs on established benchmarks, specifically, 1.57/1.39 (NFE=1/2) for CIFAR-10, 1.47/1.17 for ImageNet 64×64, and 2.8/1.61 for AFHQ 64×64.

In addition, we make a theoretical contribution showing that while the OT-based pairing in (Pooladian et al., 2023; Tong et al., 2024) is a valid approach for reducing the attraction to the false basins, due to a fundamental course-of-dimensionality, the batch size required scales exponentially as a function of the signal dimension. Given that the latter is fairly high across various domains and the former is typically constrained by memory and compute limitations, the effectiveness of this approach is limited, as demonstrated in Figure 1.

## 2 BACKGROUND

We begin by reviewing several key denoising-based generative models, with an attempt to bring them to a common form in order to highlight the source of a sampling inefficiency that they share, and we address them in our work. The Denoising Diffusion Probability Models (DDPM) (Sohl-Dickstein et al., 2015; Ho et al., 2020), as well as Denoising Score Matching (DSM) approaches, specifically the Noise Conditional Score Network (NCSN) (Song & Ermon, 2019) use the following form of denoising loss,

$$\mathrm{argmin}_\theta \mathbb{E}_{t,q(x_1),p(x|x_1,t)}\left[\|N_\theta(x,s_t) - \nabla_x \log p(x|x_1,t)\|^2\right], \tag{1}$$

where $q(x_1)$ is the target data distribution which we are given empirically. In case of DDPM, $p(x|x_1,t) = \mathcal{N}(\sqrt{\alpha_t}x_1, (1-\alpha_t)I)$ and $s_t = t$, where $1 \leq t \leq N$ is a noise scheduling index weighted by probabilities $\propto (1-\alpha_t)$, and $\alpha_t = \prod_{i=1}^t (1-\beta_i)$ and $0 < \beta_i < 1$ are a pre-defined sequence of noise scales[1]. In this framework the network $N_\theta$ models the mean of the reverse Gaussian kernels by $p(x^{t-1}|x^t) = \mathcal{N}((x^t + \beta_t N_\theta(x^t,t))/\sqrt{1-\beta_t}, \beta_t I)$, which are designed to start their operation from a source distribution, $x^N \sim p_0 = \mathcal{N}(0,I)$. In the NCSN, $p(x|x_1,t) = \mathcal{N}(x_1, \sigma_t^2 I)$ and $s_t = \sigma_t$, where $\{\sigma_t\}_{t=1}^N$ are positive noise scales, weighted $\propto \sigma_t^2$. In this approach, the network $N_\theta$

---

[1] The $\alpha_t$ defined here correspond to the $\bar{\alpha}_t$ in the derivation of Ho et al. (2020).

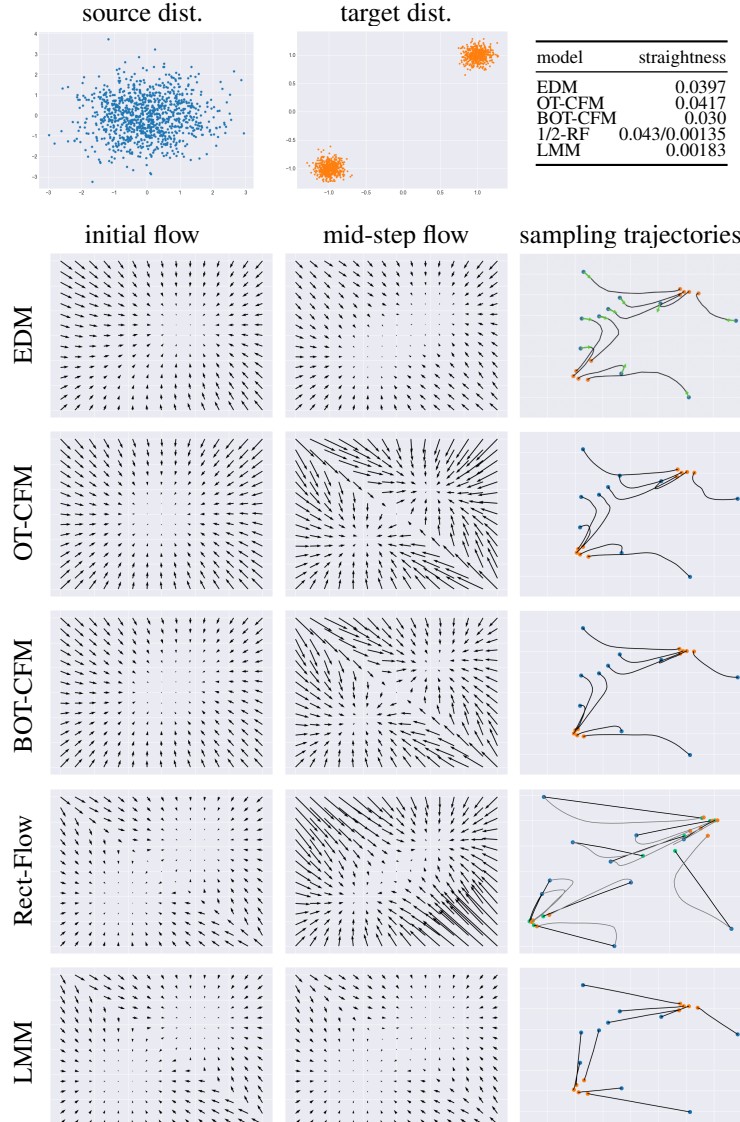

Table 1: Flow fields and sampling trajectories of different models. Top row shows the source and target distributions along their first two dimensions. The source (blue dots) is a normal distribution in 128 space dimension. The target (orange dots) is a mixture of two Gaussians located in $(-1, -1, \vec{0})$ and $(1, 1, \vec{0})$ with STDs of $(0, 1, 0.1, \vec{1})$, i.e., two separate Gaussians in the first two dimensions shown in the figure, and a normal Gaussian in the remaining 126 dimensions. The following three rows show results of the optimized DSM approach of EDM (Karras et al., 2022), the OT-CFM (Lipman et al., 2023) and its mini-batch optimized BOT-CFM (Pooladian et al., 2023) which all appear to produce curved trajectories, with an improvement observed in the BOT-CFM when pairing batches of size 256. Nearly identical results are obtained using a batch size of 128, differing in straightness by only 0.0017. The trajectories of the 1-Rect-Flow (Liu et al., 2023), shown in gray in the next row, also appear curved. The 2-Rect-Flow trajectories (black) are considerably straighter than any of the above. However, a discrepancy between these two iterations can be seen in their (target) endpoints (orange and cyan dots). This may indicate a drift from the original distribution $q$. Our LMM produces straight curves and flow Fields which are close to being constant in time. Note that excluding 2-Rect-Flow and LMM, the initial flow fields of all the methods show a clear basin of attraction at $(0, 0, \vec{0})$ responsible for an undesired drift at the beginning of the trajectories towards this point. This effect is illustrated in the EDM, where the green arrows represent the tangent vectors to the curves at their initial step. Top-right table reports the average trajectory straightness score, $\int_0^1 \|\dot{x}(t) - (x_1 - x_0)\| dt$, of each method where both 2-Rect-Flow and LMM standout.

models the score field of noised data densities $p(x, \sigma_t) = q * \mathcal{N}(0, \sigma_t^2 I)$, which is used for gradually denoising samples, starting from $x^N \sim p_0 = \mathcal{N}(0, \sigma_N^2 I)$, where $\sigma_N^2 >> \mathbb{V}[x_1]$. Much has been discussed about the close relation between these two approaches (Vincent, 2011; Song et al., 2021b; Karras et al., 2022). This formalism can be further generalized to cover continuous-time Stochastic Diffusion Equations (SDEs), where the DDPM results in a Variance Preserving (VP) process, and the NCSN in a Variance Exploding (VE) process, see (Song et al., 2021b).

The noised to clean signal regression problem solved in Eq. 1 is known to *underestimate* the true regression (Kendall & Stuart, 1973; Clarke & Gorder, 2013), due to averaging caused by the noise present in $p(x|x_1, t)$. At the limit of low Signal-to-Noise Ratio (SNR), i.e., high noise level $\sigma_t$ and $\beta_t$ (low $\alpha_t$) at large $t$ in Eq. 1, where $p(x|x_1, t) \approx p_0(x)$, the regression collapses to a constant prediction, specifically $N(x, N) \approx \mathbb{E}_{p_0}[x_1] = \mu_{p_0} = 0$ in the DDPM, and $N(x, \sigma_N) \approx \mathbb{E}_q[x_1] = \mu_q$ in the NCSN, as shown in Appendix A.1. Consequently, rather than moving towards particular instances in the target distribution, the initial sampling steps appear either stationary in the case of DDPM, or gravitate towards $\mu_q$ in the NCSN, as indicated by the green arrows in Figure 1. These instance-independent basins of attraction create inefficient sampling trajectories that lack constancy in speed and direction as also shown in the figure. The less constant in speed or direction these trajectories are, the more integration steps are needed to follow them during sampling.

Indeed, rectifying the trajectories towards fixed-speed straight lines, is an important design principle shared by recent flow-based models. The Conditional Flow Matching (CFM) method in (Lipman et al., 2023), constructs a deterministic time-dependent change-of-variable $\psi(x, t)$ that gradually maps the source distribution $p_0$ to the target $q$. Similarly to the way tractable reverse diffusion kernels are derived in DDPM (Sohl-Dickstein et al., 2015), these maps are constructed by defining simpler conditional maps $\psi_{x_1}(x, t) = (1 - t)x + tx_1$ that map $p_0$ (a normalized Gaussian), towards a small Gaussian[2] centered around each $x_1$ as a function of $t \in [0, 1]$. The network $N$ is then trained to match an aggregated velocity flow field by marginalizing $\partial \psi_{x_1} / \partial t$ over all the data points $x_1$ by solving,

$$\mathrm{argmin}_\theta \mathbb{E}_{t, q(x_1), p(x_0)}\Big[\|N_\theta\big(tx_1 + (1 - t)x_0, t\big) - (x_1 - x_0)\|^2\Big], \qquad (2)$$

As in Eq. 1 above, the network is regressed under severe marginalization where the mapping of every $x_0$ to every $x_1$ are averaged together at a non-trivial contribution at low values of $t$. In Appendix A.1 we show that, similarly to the NCSN, this approach also results in $N_\theta(x, 0) = \mu_1$ and curved trajectories, shown in Figure 1. An alternative derivation in (Albergo & Vanden-Eijnden, 2023a) discusses the option of optimizing the transport of their maps and proposes an initial direction to shorten their path length.

A flow rectification process described in (Liu et al., 2023) also matches the flow using Eq. 2, however it operates iteratively; at each step $k$ it trains $N^k$ over a different set of source $Z_0^k$ and target $Z_1^k$ examples. The process starts with the *random* pairing used in (Lipman et al., 2023), i.e., $Z_0^1$ and $Z_1^1$ are independent samples from $p_0$ and $q$ respectively. In the following steps, $Z_0^{k+1}$ and $Z_1^{k+1}$ are produced by generating new samples using $N^k$ starting from $p_0$ and $q$ (by integrating $-N^k$). This results in a *deterministic* pairing and this process is shown to monotonically increase the straightness of the trajectories in $N^k$.

As shown in Figure 1, the resulting flow trajectories at $k = 1$ share a similar gravitation towards $\mu_1$ as in the CFM. At $k = 2$ they become significantly more straight and easier to integrate. As $k$ increases errors in the estimated flow field $N^k$ accumulate and cause $Z_0^k$ and $Z_1^k$ to drift away from $p_0$ and $q$ respectively. 2-Rect-Flow ($k = 2$) is said to be found optimal in (Liu et al., 2023).

For completeness, let us note that that deterministic probability flow ODE models were also derived in the contexts of DDPM and DSM. Specifically, the Denoising Diffusion Implicit Model (DDIM) (Song et al., 2021a) derives a non-Markovian process, where the inverse kernels map noised samples along deterministic lines with noise-free endpoints. In connection to neural ODEs (Chen et al., 2018), it is shown in (Song et al., 2021b) that the DSM with Langevin Dynamics (SMLD), which is an SDE, has a deterministic time-reversal process, and following several design improvements this deterministic procedure achieve impressive results in (Karras et al., 2022). The straightness of the trajectories is not explicitly considered in these works.

---

[2]To simplify derivation we assume a zero width target Gaussian around each data point, i.e., $\sigma_{\min} = 0$ in the formalism of (Lipman et al., 2023)

**Variance Reduction.** The non-negligible association between *every* pair of samples $x_0 \sim p_0$ and $x_1 \sim q$ when marginalizing the regression losses over an independent distribution $p_0(x_0)q(x_1)$, is a common thread shared by all the models mentioned above, which undermines their sampling efficiency. As a remedy, recent works aim to replace this arbitrary pairing with ones that improve sampling.

By linking the flow's transport optimality with the straightness of their trajectories, both (Pooladian et al., 2023) and (Tong et al., 2024) derive the pairing between $p_0$ and $q$ from an Optimal Transport (OT) objective. Due to the cubic complexity of this problem (Flamary et al., 2021) (or a quadratic approximation (Altschuler et al., 2017)) the pairing, or plan $j_i$, is computed within batches of samples $\{x_0^i\}_{i=1}^n \sim p_0$ and $\{x_1^i\}_{i=1}^n \sim q$ of moderate sizes ($n = 50/256$ in (Pooladian et al., 2023)), and Eq. 2 is minimized over these permuted pairs. As shown in Figure 1 this approach results in flows that less curved than those produced by (Lipman et al., 2023). Indeed, a 30% to 60% reduction in sampling cost is reported in (Pooladian et al., 2023).

A well-known manifestation of the curse-of-dimensionality causes the ratio between the farthest and closest points to converge to a constant as the space dimension increases (Beyer et al., 1999). Thus, considerably larger batches are needed in order to find meaningful plans $\pi_{ij}$. In Appendix A.2 we provide an asymptotic analysis showing an exponential batch size $n$ dependency over the space dimension $d$ when solving a rather simpler problem; transporting a unit Gaussian distribution to itself. This finding undermines the prospect of accelerating sampling by increasing the batch size and relying solely on OT pairing. Indeed, in the example shown in Figure 1, a negligible difference in trajectory straightness is found between $n = 128$ and $n = 256$.

Another strategy to avoid independent pairing described in (Lee et al., 2023) draws $x_0 \sim q(x_0|x_1)$ given $x_1 \sim q$, where $q$ is a VAE-based encoder that maps points $x_1$ to Gaussians. In order to obtain non-trivial pairing one would seek highly distinctive $q(x_0|x_1)$ for each $x_1$ however, similarly to VAE training, Gaussians of different $x_1$ are trained to match the same $p_0$ in order to be consistent with sampling time.

## 3 LINES MATCHING MODELS

As discussed above various diffusion, score and flow matching models achieve a remarkable sampling accuracy in various data domains. This comes at a cost of executing multiple sampling steps at inference time—a notable drawback compared to the single feed-forward execution of a VAE and GAN networks. The inefficiency is rooted in the unfocused association between $p_0$ and $q$ produced by the independent example pairing, leading to poorly-resolved denoising, score and flow regression problems in the low SNR regime. Unlike methods that distill these inefficient curved trajectories (Salimans & Ho, 2022; Song et al., 2023; Kim et al., 2024), we only utilize the pairing they induce between the source and target distributions to construct a new probability flow model that matches *globally straight* lines connecting the two distributions.

We derive the Lines Matching Model (LMM) in accordance with the VE probability flow ODE formulation used in (Karras et al., 2022), by training a neural model $N_\theta$ to minimize

$$\mathcal{L}_{\text{lines}} = \mathbb{E}_{\sigma, \delta(x_1, \psi^*(x_0)), p_0(x_0)} \Big[ \| N_\theta(x_1 + \sigma x_0, \sigma) - x_1 \|_{\mathcal{P}} \Big], \tag{3}$$

The pairing function $\psi^*$ is inferred from a *deterministic* ODE-based sampling procedure $x_1 = N_{\text{Sampler}}^*(x_0)$ given a pre-trained denoising network $N^*$. In our implementation we use the DSM described in (Karras et al., 2022), commonly known as Elucidating Diffusion Models (EDM), along with its multi-stepped deterministic sampling procedure $N_{\text{Sampler}}^*$ that gradually reduces the noise level $\sigma$ in $x_1 + \sigma_{\max}x_0 \approx \sigma_{\max}x_0$, down to a negligible level where $x_1 + \sigma_{\min}x_0 \approx x_1$ (details in Appendix A.4). Let us discuss the desirable properties of the LMM, and further develop it.

**Unambiguous Pairing.** As elaborated in the previous section, training that ties every $x_0 \sim p_0$ with every $x_1 \sim q$ by conditioning the models on $x_1$ and marginalizing over this variable leads to unwanted detours in the flow map trajectories. The deterministic pairing we use, $x_1 = N_{\text{Sampler}}^*(x_0)$ for every $x_0 \sim p_0$, corresponds to example pairs $x_0, x_1$ that sample an implicit change-of-variable function $x_1 = \psi^*(x_0)$ induced by $N_{\text{Sampler}}^*(x_0)$ and $N^*$. Thus, given a state-of-the-art $N^*$ generating samples of superior quality, the mapped distribution can be considered as a good approximation,

$p_{N^*_{\text{Sampler}}} \approx q$, in this respect. Consequently, Eq. 3 regresses $N_\theta$ under a well-defined and *unambiguous pairing* between the source and target distributions *regardless* of the severity of the noise level $\sigma$.

**Globally Straight Trajectories.** Assuming $N_\theta$ is sufficiently expressive, and it satisfies Eq. 3 sufficiently well, then the lines $x_1 + \sigma_t x_0$ corresponds to its iso-contours. Thus, $N_\theta$ encodes globally straight probability flow lines connecting the source and target distributions. As noted above, while certain constructions of conditional flow maps may consist of globally straight flows, this property is lost once they are marginalized over $x_1$. In general, Eq. 3 does not pose conflicting objectives that need to be resolved.

An exception to this claim, is the availability of training pairs $x_0, N^*_{\text{Sampler}}(x_0)$ and $x'_0, N^*_{\text{Sampler}}(x'_0)$ whose connecting segments do intersect and at the same time $t$, i.e., $(1-t)x_0 + tN^*_{\text{Sampler}}(x_0) = (1-t)x'_0 + tN^*_{\text{Sampler}}(x'_0)$. In this case the regression in Eq. 3 is likely to result in an compromised intermediate solution. Such a scenario is expected to undermine both the quality of the output samples $N_\theta$ produces, and its ability to maintain a straight iso-lines. It is important to note however, that the training examples $x_0, N^*_{\text{Sampler}}(x_0)$ used in Eq. 3 correspond to solutions of a well-defined ODEs and, as discussed in (Liu et al., 2023), this implies that they are connected by non-intersecting smooth flow trajectories of $N*$.

Furthermore, the evaluation presented in Appendix A.3 demonstrates that the flows generated by the LMM maintain a very high degree of straightness, where only negligible improvement is made when applying more than 2 sampling steps. In addition, as shown in Section 4 the samples it produces at these small NFEs receive state-of-the-art FID scores.

Indeed this finding has motivated us to concentrate our efforts on improving the quality of the samples produced, i.e., the end-points of the lines, rather than their straightness by considering domain-specific metrics, adversarial loss, as well as fine-tuning $N_\theta$ to the low NFE sampling schemes used in practice, as we describe below.

**Constant-Speed Parameterization.** As noted above, constructing flows with trajectories of constant direction and *speed* is the key for efficient sampling, thanks to the trivialization of their integration. The noise scheduling commonly used in DDPM is known for its small progress (denoising) during its initial phase, as noted in (Liu et al., 2023; Lipman et al., 2023), and exemplifies the need for fixed speed. By training $N_\theta$ in Eq. 3 to match endpoints, $x_1$, rather than denoising vector fields, we obtain constancy in speed by design. Specifically, at every point along the segment $x_\sigma = x_1 + \sigma x_0$, the *remaining* path towards $x_1$, is given by

$$v(x_\sigma, \sigma) = N_\theta(x_\sigma, \sigma) - x_\sigma. \tag{4}$$

Hence, a constant speed parameterization with respect to $\sigma$ is given by $v/\sigma$ in the VE formulation that we follow. This can be used it to derive the discretization of an arbitrary number of steps in which a uniform progression along the lines is made.

**Domain-Specific Loss.** Another benefit of matching $x_1$ in Eq. 3, rather than flow vectors, such as $x_1 - x_0$ as done in (Lipman et al., 2023; Liu et al., 2023), allows us to exploit the fact that $N_\theta$ matches native signals, and hence domain-specific metrics can be employed. In particular, this allows us to use the perceptual loss in (Johnson et al., 2016) to define $\|\cdot\|_{\mathcal{P}}$, when $q$ corresponds to a distribution of images. Indeed in Appendix A.3 we compare the use of this metric to $L_2$ loss, and show a substantial improvement in sampling fidelity lowering the FID over the CIFAR-10 dataset from 5.125 to 3.124 (NFE=1), and from 4.289 to 2.796 (NFE=2).

**Adversarial Loss.** Eq. 3 trains $N_\theta$ to replicate samples $x_1$ generated by $N^*_{\text{Sampler}}(x_0)$, rather than being trained directly on authentic (input) samples from $q$. This sets a limit over the quality at which $N_\theta$ approximates $q$—one which is bounded by the quality of the mediator network $N^*$ and its sampling procedure, $N^*_{\text{Sampler}}$. Training $N_\theta$ to produce signals in their original domain, e.g., clean images, in Eq. 3, offers yet another advantage; we can follow the strategy of (Kim et al., 2024) and bootstrap $N_\theta$ to the original training data using an adversarial loss. Specifically, we train a discriminator network $D$ to discriminate between authentic training samples $x_1 \sim q$ and ones produced by $N_\theta$, by

$$\mathcal{L}_{\text{disc}} = \mathbb{E}_{\sigma, \delta(x_1, \psi^*(x_0)), p_0(x_0)} \Big[ \log\big(1 - D\big(N_\theta\big(x_1 + \sigma x_0, \sigma\big)\big)\big) \Big] + \mathbb{E}_{q(x_1)} \Big[ \log\big(D(x_1)\big) \Big], \tag{5}$$

where we use the architecture in (Sauer et al., 2022) and the adaptive weighing $\lambda_{\text{adapt}}$ in (Esser et al., 2021) that Kim et al. use. We finally train $N_\theta$ to minimizing $\lambda_{\text{lines}}\mathcal{L}_{\text{lines}} + \lambda_{\text{adapt}}\mathcal{L}_{\text{disc}}$. We provide all the implementation details in Appendix A.4.

As we show in Appendix A.3, training the LMM without $\mathcal{L}_{\text{disc}}$ achieves fairly satisfying FID scores, namely 3.12 (NFE=1) and 2.79 (NEF=2) on CIFAR-10. By incorporating the latter these scores further improve to 1.67 (NFE=1) and 1.39 (NFE=2). This *surpasses* the quality of samples generated by $N^*_{\text{Sampler}}(x_0)$ which uses more sampling steps (NFE=35) and achieves FID of 1.79.

**Sampling-Optimized Training (SOT).** Motivated by the evaluation reported in Appendix A.3, showing that the LMM achieves its high-quality samples already at NFE $\leq$ 3, we explored the option of further improving its performance by restricting the training of $N_\theta$ to the specific steps (noise levels $\sigma$) used at sampling time. Appendix A.3 shows the further quality increase, from FID 1.67 to 1.57 (NFE=1), thanks to this training strategy.

# 4 EVALUATION AND COMPARISON

We trained the LMM on three benchmark datasets, CIFAR-10, ImageNet 64×64, and AFHQ 64×64, which are commonly used for evaluating generative models. We used the same network architecture and hyper-parameters as existing models, with all the implementation details provided in Appendix A.4.

**Quantitative Comparison.** Table 2 provides a comprehensive comparison of the CIFAR-10 reproduction quality achieved by different models. The comparison clearly shows that diffusion-based models achieve lower FID scores, albeit at an increased sampling cost compared to GANs. Flow-matching models demonstrate their ability to reduce the NFEs, alongside a range of distillation techniques that operate effectively with very low NFEs—one or two sampling steps.

Among these, the Consistency Trajectory Model (CTM) (Kim et al., 2024), achieves excellent FID scores of 1.73 (NFE=1) and 1.63 (NFE=2) on conditional CIFAR-10. Our LMM surpasses these scores and sets new state-of-the-art scores of 1.57 and 1.39 respectively. We note that both methods benefit from the use of an adversarial loss, but as reported in Section A.3, the LMM's performance remains better also without this loss. We attribute this to the fact that the LMM produces favorable line flow trajectories, rather than relying on the curved EDM trajectories that the CTM distills.

The Rect-Flow in (Liu et al., 2023) achieves an impressive FID score of 4.85 in its second iteration, where it produced significantly straighter trajectories (as shown in Figure 1). We note that this second iteration achieves the best trade-off between straightness and errors that this scheme accumulates.

Table 4 shows the results obtained on a larger dataset, ImageNet 64×64. Here too, the LMM demonstrates state-of-the-art performance, with a notable improvement at NFE=2, where it reaches an FID of 1.17. The SiD Zhou et al. (2024) trains a single-step generator to agree with a pre-trained EDM, achieving an impressive score of 1.52. Unlike the LMM, SiD does not rely on the EDM to generate training examples; instead, it uses it to define the generator's loss while simultaneously training an additional score-matching network. This approach poses significantly higher GPU memory requirements and operations during training.

Finally, Table 5 reports the results on the AFHQ 64×64 dataset, where the LMM shows lower FID scores using significantly fewer NFEs compared to the EDM despite the fact that the latter is used to produce the initial correspondence between $p_0$ and $q$. This is also the case in Tables 2 and 4. While achieving a state-of-the-art FID of 1.54 at NFE=2, the SiD achieves a better score using a single step. We note that unlike the CIFAR-10 and ImageNet 64×64cases, the discriminator architecture and hyper-parameters we used were not we used were not tailored to this dataset in previous work (e.g., StyleGAN-XL Sauer et al. (2022)). This affected the expected improvements from the SOT strategy, as discussed in Appendix A.3, and we therefore believe the LMM has greater potential on this dataset.

In terms of Inception Score (IS), the LMM achieves state-of-the-art results, scoring above 10 for both NFEs on CIFAR-10, as shown in Table 2. On ImageNet 64×64, the LMM improves upon its teacher model (EDM), although StyleGAN-XL attains the highest score. Among diffusion-based

Table 2: CIFAR-10

| Model | NFE | unconditional | | conditional |
|---|---|---|---|---|
| | | FID | IS | FID |
| **GAN** | | | | |
| BigGAN Brock et al. (2019) | 1 | 14.70 | 9.22 | - |
| StyleGAN2-ADA Karras et al. (2020) | 1 | 2.92 | 9.83 | 2.42 |
| StyleGAN-D2D Kang et al. (2024) | 1 | - | - | 2.26 |
| StyleGAN-XL Sauer et al. (2022) | 1 | - | - | 1.85 |
| **Diffusion / Score Matching** | | | | |
| DDPM Ho et al. (2020) | 1000 | 3.17 | 9.46 | - |
| DDIM Song et al. (2021a) | 100 | 4.16 | - | - |
| Score SDE Song et al. (2021b) | 2000 | 2.20 | 9.89 | - |
| EDM Karras et al. (2022) | 35 | 1.97 | 9.84 | 1.79 |
| **Distillation / Direct Gen.** | | | | |
| KD Luhman & Luhman (2021) | 1 | 9.36 | 8.36 | - |
| PD Salimans & Ho (2022) | 1 | 9.12 | - | - |
| CT Song et al. (2023) | 1 | 8.70 | 8.49 | - |
| CD Song et al. (2023) | 1 | 3.55 | 9.48 | - |
| CD+GAN Lu et al. (2023) | 1 | 2.65 | - | - |
| iCT Song & Dhariwal (2024) | 1 | 2.83 | 9.54 | - |
| iCT-deep Song & Dhariwal (2024) | 1 | 2.51 | 9.76 | - |
| CTM Kim et al. (2024) | 1 | 1.98 | - | 1.73 |
| DMD Yin et al. (2024) | 1 | 3.77 | - | 2.66 |
| SiD ($\alpha = 1$) Zhou et al. (2024) | 1 | 2.02 | 10.02 | 1.93 |
| SiD ($\alpha = 1.2$) Zhou et al. (2024) | 1 | 1.92 | 9.98 | 1.71 |
| PD Salimans & Ho (2022) | 2 | 4.51 | - | - |
| CT Song et al. (2023) | 2 | 5.83 | 8.85 | - |
| CD Song et al. (2023) | 2 | 2.93 | 9.75 | - |
| iCT Song & Dhariwal (2024) | 2 | 2.46 | 9.80 | - |
| iCT-deep Song & Dhariwal (2024) | 2 | 2.24 | 9.89 | - |
| CTM Kim et al. (2024) | 2 | 1.87 | - | 1.63 |
| **Flow Matching** | | | | |
| OT-CFM Lipman et al. (2023) | 142 | 6.35 | - | - |
| 1-Rect-Flow (distill) Liu et al. (2023) | 1 | 6.18 | 9.08 | - |
| 2-Rect-Flow (distill) Liu et al. (2023) | 1 | 4.85 | 9.01 | - |
| 3-Rect-Flow (distill) Liu et al. (2023) | 1 | 5.21 | 8.79 | - |
| 1-Rect-Flow Liu et al. (2023) | 127 | 2.58 | 9.60 | - |
| 2-Rect-Flow Liu et al. (2023) | 110 | 3.36 | 9.24 | - |
| 2-Rect-Flow Liu et al. (2023) | 104 | 3.96 | 9.01 | - |
| LMM | 1 | **1.90** | **10.16** | **1.57** |
| LMM | 2 | **1.55** | **10.20** | **1.39** |

Table 4: ImageNet 64×64

| Model | NFE | conditional | |
|---|---|---|---|
| | | FID | IS |
| **GANs** | | | |
| BigGAN-deep Brock et al. (2019) | 1 | 4.06 | - |
| StyleGAN-XL Sauer et al. (2022) | 1 | 1.51 | **82.35** |
| **Diffusion / Score Matching** | | | |
| RIN Jabri et al. (2023) | 1000 | 1.23 | - |
| EDM Karras et al. (2022) | 511 | 1.36 | - |
| DDPM Ho et al. (2020) | 250 | 11 | - |
| EDM Karras et al. (2022) | 79 | 2.23 | 48.88 |
| **Distillation / Direct Gen.** | | | |
| PD Salimans & Ho (2022) | 1 | 15.39 | - |
| BOOT Gu et al. (2023) | 1 | 16.30 | - |
| CT Song et al. (2023) | 1 | 13.0 | - |
| CD Song et al. (2023) | 1 | 6.20 | 40.08 |
| iCT Song & Dhariwal (2024) | 1 | 4.02 | - |
| iCT-deep Song & Dhariwal (2024) | 1 | 3.25 | - |
| CTM Kim et al. (2024) | 1 | 1.92 | 70.38 |
| DMD Yin et al. (2024) | 1 | 2.62 | - |
| SiD ($\alpha = 1$) Zhou et al. (2024) | 1 | 2.02 | - |
| SiD ($\alpha = 1.2$) Zhou et al. (2024) | 1 | 1.52 | - |
| PD Salimans & Ho (2022) | 2 | 8.95 | - |
| CT Song et al. (2023) | 2 | 11.1 | - |
| CD Song et al. (2023) | 2 | 4.70 | - |
| iCT Song & Dhariwal (2024) | 2 | 3.20 | - |
| iCT-deep Song & Dhariwal (2024) | 2 | 2.77 | - |
| CTM Kim et al. (2024) | 2 | 1.73 | 64.29 |
| **Flow Matching** | | | |
| OT-CFM Lipman et al. (2023) | 138 | 14.45 | - |
| BOT-CFM Pooladian et al. (2023) | 132 | 11.82 | - |
| LMM | 1 | **1.47** | 59.86 |
| LMM | 2 | **1.17** | 61.18 |

Table 5: AFHQ 64×64

| Model | NFE | FID |
|---|---|---|
| **Score Matching** | | |
| EDM Karras et al. (2022) | 79 | 1.96 |
| **Distillation** | | |
| SiD ($\alpha = 1.2$) Zhou et al. (2024) | 1 | 1.71 |
| SiD ($\alpha = 1$) Zhou et al. (2024) | 1 | **1.63** |
| LMM | 1 | 2.68 |
| LMM | 2 | **1.54** |

Table 3: ImageNet 64×64 Samples Comparison.

models, the LMM receives an IS of 61.18 using 2 NFEs, which is closely competitive with CTM which scores 64.29.

**Visual Evaluation.** Table 3 compares samples produced by the EDM and LMM with and without an Adversarial Loss (ADL), and using 1 or 2 sampling steps. Incorporating the ADL appears to be related to fine image details, contributing to their richness and resolvedness. This is observed in the fish background, lettuce leaves, the bird feathers, and the man's face. The second sampling iteration (NFE=2 in the table) has a larger scale impact, improving the correctness of the objects' shape, as well as the consistency between different objects. This effect can be seen in the clerk's body and face, the bird's body, the shape of the bread/cake, and the matching red shoes. The images generated by the EDM generally appear to be less detailed, although there are clear exceptions to that. A similar comparison of CIFAR-10 and AFHQ 64×64 is shown in Table 10.

Additional example samples produced by the LMM on each of the datasets are shown in Figures 1, 2, and 3.

## 5 CONCLUSIONS

In this work, we showed that using broad random correspondences between the source and target distributions results in collapsed predictions at low SNRs. By bringing them to a common analytical framework, we showed that this degeneracy is inherent to key models, including denoising diffusion, score-matching, and flow-matching techniques. We used this insight to propose a solution by deriving a deterministic correspondence from ODE-based sampling. To avoid the inefficiencies in the resulting trajectories, we only used their endpoints to train our LMM which parameterizes the transition between distributions using globally straight lines.

We leveraged the fact that our formulation works directly on signal reconstruction, and proposed several training losses and strategies to improve the quality of the generated samples. In doing so, our work bridges the domains of flow matching and denoising distillation. The combined effect of enhanced sampling quality and sampling efficiency has enabled the LMM to achieve state-of-the-art image generation quality in just one or two sampling steps.

Finally, as part of our effort to understand and improve the pairing required for training flow models, we made a theoretical contribution showing that OT-based pairing suffers from an exponential relationship between the size of the paired sets (mini-batches) and the signal dimension.

Our work leaves one important goal unaddressed: avoiding the reliance on a pre-trained model, and establishing its pairing in an ab initio manner. As a future research direction we intend to investigate the adaptation of an iterative scheme, like the one in Liu et al. (2023), while avoiding drifts in the training data during this process.

**Code Reproducibility Statement.** In Appendix A.4 we provide detailed information on the network architecture and hyper-parameters used to produce the reported results. Additionally, we plan to publicly release our code and the trained LMM network weights.

**Social Impact Statement.** Given their increasing prevalence, improving the efficiency of generative AI models is likely to result in a significant reduction in computational costs and energy usage. However, we are fully aware of the risks associated with these models and wish to express our strong opposition to any unethical use.

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

# A  APPENDIX

## A.1  REGRESSION AT LOW SIGNAL-TO-NOISE RATIOS

Both DDPM (Sohl-Dickstein et al., 2015; Ho et al., 2020) and DSM, e.g., NCSN (Song & Ermon, 2019) and EDM (Karras et al., 2022), start their sampling process from an easy-to-sample source distribution $x^N \sim p_0$, typically a Gaussian. Hence their denoising networks $N_\theta$ are trained to operate on these distributions. At $t = N$, Eq. 1 becomes

$$\mathrm{argmin}_\theta \mathbb{E}_{q(x_1), p(x|x_1, N)} \Big[ \|N_\theta(x, s_N) - \nabla_x \log p(x|x_1, N)\|^2 \Big]. \tag{6}$$

In the case of NCSN, $s_N = \sigma_N$ and $p(x|x_1, \sigma_N) = \mathcal{N}(x_1, \sigma_N^2 I)$ where $\sigma_N^2 >> \mathbb{V}[x_1]$, i.e., a very low SNR, allowing to be approximate this distribution by a pure noise at sampling time, specifically $p_0 = \mathcal{N}(0, \sigma_N^2 I)$.

Noting that $\nabla_x \log p(x|x_1, N) = (x_1 - x)/\sigma_N^2$, Eq. 6 becomes (at $t = N$),

$$\mathrm{argmin}_\theta \mathbb{E}_{q(x_1), x_0 \sim p_0} \Big[ \|N_\theta(x_0, \sigma_N) - (x_1 - x_0)/\sigma_N^2\|^2 \Big], \tag{7}$$

where every $x_0 \sim p_0$ is equally regressed to match every $x_1 \sim q$. Regressing under such indeterminacy is poised to result in the degenerate averaged prediction $N_\theta(x, \sigma_N) = (\mu_q - x)\sigma_N^{-2}$. This topic is thoroughly discussed in (Kendall & Stuart, 1973; Clarke & Gorder, 2013). Finally, we note that at sampling stage the factor $\sigma_N^{-2}$ is typically canceled by using time steps proportional to $\sigma_N^2$, see for example (Song & Ermon, 2019) and (Karras et al., 2022). Thus, the sampling trajectories are drawn towards $\mu_q$, up to some implementation-dependent speed factors, during their first steps. This effect is highlighted by the green arrows in Figure 1.

Analogously, the DDPM noise scheduling is set such that $\alpha_N$ is small, e.g., $\alpha_N = 6 \times 10^{-3}$ in (Ho et al., 2020) and $\alpha_N = 5 \times 10^{-5}$ in (Nichol & Dhariwal, 2021). Therefore $p(x|x_1, N) = \mathcal{N}(\sqrt{\alpha_N} x_1, (1 - \alpha_N)I) \approx \mathcal{N}(0, I)$ which, here as well, can be replaced with the source distribution $p_0$ during sampling. In this case, Eq. 6 becomes (again, at $t = N$),

$$\mathrm{argmin}_\theta \mathbb{E}_{q(x_1), x_0 \sim p_0} \Big[ \|N_\theta(x_0, N) - (\sqrt{\alpha_N} x_1 - x_0)/(1 - \alpha_N)\|^2 \Big], \tag{8}$$

resulting in $N_\theta(x, N) = (\sqrt{\alpha_N} \mu_1 - x)/(1 - \alpha_N)$. In fact, this should be interpreted as $N_\theta(x, N) = (\sqrt{\alpha_N} \mu_1 + \sqrt{(1 - \alpha_N)} \mu_0 - x)/(1 - \alpha_N) \approx \mu_0 - x$, since $\alpha_N << 1$ and we added the term $\sqrt{(1 - \alpha_N)} \mu_0$ since $\mu_0 = 0$. More fundamentally, the DDPM noising process $p(x|x_1, N) = \mathcal{N}(\sqrt{\alpha_t} x_1, (1 - \alpha_t)I)$ gradually replaces every data sample $x_1$ with a normal Gaussian by shifting the mean from $x_1$ towards $\mu_0$ (chosen to be 0 for convenience) and by increasing the variance from $(1 - \alpha_1) \approx 0$ to $\mathbb{V}[x_0] = 1$.

Thus, at the $N$-th step of the DDPM sampling step, the denoiser collapses to the mean of the source distribution. Consequently, its flow trajectories gravitate toward $\mu_0 = 0$ during their earlier steps. This affects only the magnitude of initial (full noise) states, and the sample's shape evolves only in later steps, thus the DDPM sampling process is often described as stagnant during its early stages (e.g., in Figure 6 in (Lipman et al., 2023)).

Finally, the flow models in (Lipman et al., 2023), and (Liu et al., 2023) at 1-Rect-Flow, regress arbitrary samples from $p_0$ to the data points $x_1$ at time $t = 0$, where its training loss, Eq. 9, becomes

$$\mathrm{argmin}_\theta \mathbb{E}_{q(x_1), p(x_0)} \Big[ \|N_\theta(x_0, 0) - (x_1 - x_0)\|^2 \Big], \tag{9}$$

Similarly to Eq. 7, also Eq. 9 regresses points $x_0$ with direction towards arbitrary data points, $x_1 - x_0$. This leads again to a degenerate solution where $N_\theta(x_0, 0) = \mu_1 - x_0$, which similarly to the score-matching approach, biases the sampling trajectories towards $\mu_1$ at their earlier stages. This effect is also observed in Figure 1.

## A.2 BATCH OPTIMAL TRANSPORT - BATCH SIZE ANALYSIS

We assess here the asymptotic dependence in the Batch OT CFM (BOT-CFM) methods described in (Pooladian et al., 2023) and (Tong et al., 2024) over the batch size $n$ as a function of space dimension $d$. In these works, following the notations of the former, the independent distribution $p_0(x_0)q(x_1)$ in Eq. 2 is replaced by a joint distribution $q(x_0, x_1)$ induced by a batch-optimized coupling, $\{x_0^i\}_{i=1}^n \sim p_0$ and $\{x_1^{j_i}\}_{i=1}^n \sim q$, where the permutation $j_i$ optimizes the transport cost $\|x_0^i - x_1^{j_i}\|^2$ within each batch. Combining this with the OT conditional flow map $\psi_{x_1}(x, t)$ in (Lipman et al., 2023), the BOT-CFM training loss is given by

$$\mathrm{argmin}_\theta \mathbb{E}_{t, \{x_0^i\}_{j=1}^n \sim p_0, \{x_1^i\}_{i=1}^n \sim q}\Big[\sum_{i=1}^n \|N_\theta\big((1-t)x_0^i + tx_1^{j_i}, t\big) - \big(x_1^{j_i} - x_0^i\big)\|^2\Big], \quad (10)$$

To simplify the analysis we consider a fairly naive problem of finding a mapping from a normal Gaussian in $\mathbb{R}^d$ to itself, where the optimal solution is given by the identity mapping. In the context of matching the velocity field, as done in (Pooladian et al., 2023; Tong et al., 2024), the optimal field is given by $N_\theta(x, t) = 0$. As shown in Appendix A.1, in case of independent distribution $p_0(x_0)q(x_1)$ (the solution of Eq. 9)) the resulting vector field at $t = 0$ is $N_\theta(x_0, 0) = \mu_1 - x_0 = -x_0 \neq 0$ which is clearly far from the optimum.

In the BOT-CFM (at $t = 0$) closer and closer $x_1^{j_i}$ will be found to each $x_0^i$ as the batch size increases, and hence by training $N_\theta(x_0^i, 0)$ to match $x_1^{j_i} - x_0^i$, in Eq. 10, a reduced velocity vector is expected. The question of how fast this decrease takes place as a function of $d$ is critical, as only moderately sized batches can be used in practice.

We address this question at $t = 0$, where Eq. 10 simplifies to a simple regression problem over $x_0$,

$$\mathrm{argmin}_\theta \mathbb{E}_{\{x_0^i\}_{j=1}^n \sim p_0, \{x_1^i\}_{i=1}^n \sim q}\Big[\sum_{i=1}^n \|N_\theta\big(x_0^i, 0\big) - \big(x_1^{j_i} - x_0^i\big)\|^2\Big], \quad (11)$$

which is solved by,

$$N_\theta\big(x_0, 0\big) = \mathbb{E}_{p^{B_1^n}(x_1^*|x_0)}\big[x_1^* - x_0\big], \quad (12)$$

where $p^{B_1^n}(x_0, x_1^*)$ is the joint distribution induced by finding the optimal pairing between source $x_0^i$ and target $x_1^{j_i}$ within each batch $B_1^n$ of size $n$.

The case $n = 1$ (equivalent to random pairing), we get $p^{B_1^1}(x_1^*|x_0) = p_0(x_0)q(x_1)$ which was discussed above and results in a velocity $N_\theta(x_0, 0) = -x_0$ attracting sampling trajectories towards $\mu_1 = 0$ at $t = 0$, instead of remaining stationary, thus producing the unnecessarily curved trajectories. As $n$ increases, however, the chances to regress $x_0$ to closer $x_1^*$ increases and thus a shift in $\mathbb{E}_{p^{B_1^n}(x_1^*|x_0)}\big[x_1^*\big]$ toward $x_0$ is expected. In order to analyze the magnitude of this shift as a function of both $n$ and $d$, let us review basic properties of random vectors in $\mathbb{R}^d$.

Let $x$ and $y$ be two independent normal scalars drawn from $\mathcal{N}(0, 1)$. Their product $xy$ is a random variable with the following moments

$$\mathbb{E}[xy] = \mathbb{E}[x]\mathbb{E}[y] = 0 \quad (13)$$

and,

$$\mathbb{V}[xy] = \mathbb{E}\big[(xy)^2\big] = \mathbb{E}\big[x^2\big]\mathbb{E}\big[y^2\big] = \mathbb{V}[x]\mathbb{V}[y] = 1 < \infty, \quad (14)$$

both follow from the normality and independence of $x, y$. Let us consider now two independent normal vectors $x, y \in \mathbb{R}^d$, drawn from $\mathcal{N}(0, I)$, and their dot-product, defined by

$$\langle x, y \rangle = \frac{1}{d}\sum_{i=1}^d x^i y^i. \quad (15)$$

Being an average of independent random variables, at large space dimension $d$ the central limit theorem becomes applicable and provides us its limit distribution by,

$$\langle x, y \rangle \xrightarrow{d} \mathcal{N}(0, d^{-1}), \tag{16}$$

which is calculated from the scalar moments in Eq. 13 and Eq. 14. This implies that as the space dimension $d$ increases, this distribution gets more concentrated around 0, meaning that the vectors $x$ and $y$ are becoming less likely to be related to one another by becoming increasingly orthogonal. As we shall now show, this makes the task of finding $x_1 \in B_1^n$ close to $x_0$ within finite batches increasingly difficult as $d$ grows. This relates to a well-known phenomenon where the ratio between the farthest and closest points converges to a constant, as the space dimension increases (Beyer et al., 1999).

Indeed, by considering the magnitude of the regressed flow velocity in Eq. 17,

$$\left\| \mathbb{E}_{p^{B_1^n}(x_1^*|x_0)} \left[ x_1^* \right] - x_0 \right\|^2 = \left\| \mathbb{E}_{p^{B_1^n}(x_1^*|x_0)} \left[ x_1^* \right] \right\|^2 + \|x_0\|^2 - 2 \left\langle \mathbb{E}_{p^{B_1^n}(x_1^*|x_0)} \left[ x_1^* \right], x_0 \right\rangle$$
$$\geq \|x_0\|^2 - 2 \left\langle \mathbb{E}_{p^{B_1^n}(x_1^*|x_0)} \left[ x_1^* \right], x_0 \right\rangle = \|x_0\|^2 - 2 \mathbb{E}_{p^{B_1^n}(x_1^*|x_0)} \left[ \langle x_1^*, x_0 \rangle \right], \tag{17}$$

we clearly see the need for increased dot-product similarity within the batches $B_1^n$ in order to reduce the magnitude of the learned target flow velocity—ideally zero in this problem. In this derivation $\left\| \mathbb{E}_{p^{B_1^n}(x_1^*|x_0)} \left[ x_1^* \right] \right\|^2$ is neglected as we are in a process of deriving a lower bound for flow velocity field, $\left\| \mathbb{E}_{p^{B_1^n}(x_1^*|x_0)} \left[ x_1^* \right] - x_0 \right\|^2$. We also note that the last equality follows from the linearity of the dot-product operator.

As an upper bound for $\left\langle \mathbb{E}_{p^{B_1^n}(x_1^*|x_0)} \left[ x_1^* \right], x_0 \right\rangle$ we assume that this similarity is computed by pairing $x_0$ with its *closest* $x_1^* \in B_1^n$ without considering trade-offs that arise when pairing a complete batch of source points $\{x_0^i\}_{j=1}^n \sim p_0$ with the batch of target points, in $B_1^n$, as done in practice in BOT-CFM, in Eq. 10.

In this scenario, $\langle x_1^*, x_0 \rangle = \max_i \langle x_1^i, x_0 \rangle$, where $\langle x_1^i, x_0 \rangle$ are independent variables and, as shown above, $\langle x_1^i, x_0 \rangle \sim \mathcal{N}(0, d^{-1})$. Using Jensen's inequality, we get that

$$\exp\left( t \mathbb{E}_{p^{B_1^n}(x_1^*|x_0)}[\langle x_1^*, x_0 \rangle] \right) \leq \mathbb{E}_{p^{B_1^n}(x_1^*|x_0)} \left[ \exp(t\langle x_1^*, x_0 \rangle) \right] = \mathbb{E}_{\mathcal{N}(0,d^{-1})} \left[ \max_i \exp(t\langle x_1^i, x_0 \rangle) \right]$$
$$\leq \sum_{i=1}^n \mathbb{E}_{\mathcal{N}(0,d^{-1})} \left[ \exp(t\langle x_1^i, x_0 \rangle) \right] = n \exp\left( \frac{t^2}{2d} \right), \tag{18}$$

where the last equality follows from the calculation of the moment generating function of the Gaussian distribution, $\mathcal{N}(0, d^{-1})$. Thus, by taking the logarithm of Eq. 18 and dividing by $t$ we get

$$\mathbb{E}_{p^{B_1^n}(x_1^*|x_0)}[\langle x_1^*, x_0 \rangle] \leq \log(n)/t + \frac{t}{2d}. \tag{19}$$

Finally, by setting $t = \sqrt{2d \log n}$, we get

$$\mathbb{E}_{p^{B_1^n}(x_1^*|x_0)}[\langle x_1^*, x_0 \rangle] \leq \sqrt{\frac{2 \log n}{d}}. \tag{20}$$

This relation implies that in order to obtain a proper (zero) target velocity field in Eq. 17, the batch size $n$ must grow exponentially as a function of the space dimension $d$, which tends to be fairly large in practical settings. Indeed, as demonstrated in Figure 1 already at $d = 128$ the BOT-CFM shows a moderate reduction in the average trajectory straightness compared to the CFM using batch sizes of $n = 128$. The use of $s = 256$ offered a negligible improvement. We conclude that this dependence undermines the prospect of achieving additional substantial improvement over the one reported in (Pooladian et al., 2023) by increasing the batch size and relying solely on the BOT strategy.

Several notes on the scope of our analysis which considered a simple problem of mapping two Gaussians and considered the affairs at $t = 0$. First, it shows that even over an arguably simple problem the effectiveness of the BOT-CFM is limited by its asymptotic. Second, as discussed at great length in Section 2 a major source of sampling inefficiency, shared by multiple key approaches, takes

place at the vicinity of $t = 0$, and hence the focus of our analysis to this time should not necessarily be considered as a limitation. Finally, most of the arguments made above remain valid when real-world target data distribution $q$ is used. Namely, the limiting orthogonal distribution in Eq. 16 and hence the exponential batch size requirement for finding real-world data point $x_1^*$ sufficiently close to a random latent vector $x_0 \sim \mathcal{N}(0, d^{-1})$. Our restriction to a target Gaussian distribution is made specifically for the purpose of being able to consider the analytical results with respect to a *known* optimal flow field.

### A.3 ABLATION STUDIES

We report here the results of several empirical experiments that assess the impact of different components related to LMM's training, described in Section 3, on its sampling performance and quality.

**Domain-Specific versus L2 Loss.** Training the LMM to reproduce the end-points of the probability flow lines, i.e., noise-free images, allows us employ perceptual metrics, specifically (Johnson et al., 2016), for training. This loss is known to provide visually-preferable optimization trade-offs in various applications, see (Zhang et al., 2018). Table 7 shows that training the LMM using a VGG-based perceptual loss (VGG) achieves lower FID scores compared to that of L2 loss at all NFEs tested. The ability to use this reconstruction loss is inherent to the design of the LMM, and is not shared by all flow-based approaches, e.g., (Lipman et al., 2023; Liu et al., 2023).

**Number of Sampling Steps.** Tables 7, 8, and 9 report the FID scores on different datasets using different NFEs and sampling steps. Specifically, we used subsets of the sampling steps from the sampling scheme in (Karras et al., 2022). While the number of steps provides some amount of ability to trade-off between quality and efficiency, it is clear from these tables that increasing the NFEs suffers from a diminishing return. This finding aligns with the explanation that the probability flow lines generated by the LMM are fairly straight, and that the sampling errors are primarily due to the accuracy of their endpoints, i.e., the quality at which the target samples $x_1 \sim q$ can be reproduced by exact integration. This further motivated us in Section 3 to focus on improving the sample reproduction, as we evaluate next.

**Adversarial Loss.** Indeed, Tables 7, 8, and 9, show that the incorporation of an adversarial loss (ADL) provides an additional significant improvement to the image quality produced by the LMM. Indeed, this addition also helped the CTM in (Kim et al., 2024) to improve their baseline, specifically, FID of 2.28 using a discriminator and 5.19 without it, using NFE=1 on CIFAR-10. We attribute the lower FID scores achieved by the LMM, in both scenarios, to the fact that it models favorable line flow trajectories, rather than the original curved EDM's trajectories, which are distilled in (Kim et al., 2024).

**Sampling-Optimized Training.** Motivated by limited improvement higher NFEs produce, in Section 3 we proposed another strategy to improve sample quality by restricting the training to the specific time steps used at the sampling stage. Tables 7 and 8 show that this training strategy also has the ability to contribute significantly despite the fact that it adds no cost. Table 9 an opposite trend which appears to be related to a saturation (over-fitting) due to two factors: (i) to limited data available in this dataset, and (ii) the SOT focuses on high noise levels, which makes it easier to discriminate between generated and real samples. We conclude that a more fine-tuned discriminator setting is needed to achieve optimal results.

### A.4 IMPLEMENTATION DETAILS

We implemented the LMM in PyTorch and trained it on four GeForce RTX 2080 Ti GPUs on three commonly used benchmark datasets: CIFAR-10, ImageNet 64×64, and AFHQ 64×64(aka. AFHQ-v2 64×64). We employed the network architectures and hyper-parameters listed in Table 6, which were previously used in (Karras et al., 2022; Song et al., 2023; Kim et al., 2024; Lipman et al., 2023) over these datasets.

**Training Losses.** As noted above, we used the VGG-based perceptual loss in (Johnson et al., 2016) as our image reconstruction loss term in Eq. 3, and resized the images to 224-by-224 pixels before evaluating it.

We use a similar adversarial loss as the one used in (Kim et al., 2024), namely, we adopted the discriminator architecture from (Sauer et al., 2022) and used its conditional version when training on labeled datasets. We used the same feature extraction networks they use, as well as the adaptive weighing from (Esser et al., 2021), given by $\lambda_{\text{adapt}} = \|\nabla_{\theta_L} \mathcal{L}_{\text{lines}}\| / \|\nabla_{\theta_L} \mathcal{L}_{\text{disc}}\|$, where $\theta_L$ denotes the weights of the last layer of $N_\theta$. We also used their augmentation strategy, taken from (Zhao et al., 2020), we resized the images to 224-by-224 pixels before applying this loss as well.

**Denoising ODE.** As noted in Section 3, we use the EDM denoising score-matching model $N^*$ in (Karras et al., 2022) in order to produce our training pairs $x_0, N^*_{\text{Sampler}}(x_0)$. We use their deterministic sampler (second-order Heun) in order to establish a well-defined change-of-variable, $N^*_{\text{Sampler}}(x)$, between the source and target distributions. This scheme uses a source distribution $p_0 = \mathcal{N}(0, \sigma_{\max})$ and noise scheduling $\sigma_t = \left(\sigma_{\max}^{1/\rho} + t/(N-1)(\sigma_{\min}^{1/\rho} - \sigma_{\max}^{1/\rho})\right)^\rho$, where $\rho = 7$ and $\sigma_{\min} = 0.002$ which corresponds to a negligible noise level when reaching the target distribution, $q$, assuming $\mathbb{V}[x_1]$ of order around 1. This method uses $N$=18 (NFE=35) steps to draw samples from the CIFAR-10 dataset, and $N$=40 (NFE=79) for ImageNet 64×64 and AFHQ 64×64.

**Sampling the LMM.** We use the sampling scheme used in (Song et al., 2023; Kim et al., 2024) to sample the LMM. This consists of the following iterations, $x^{t+1} = N_\theta(x^t, \sigma_t) + \sigma_{t+1}\eta$, where $x^0 \sim p_0$ and $\eta \sim \mathcal{N}(0, I)$. We report the noise scheduling we use in each step, $\sigma_t$, in terms of the ones used in (Karras et al., 2022), in Tables 7, 8, and 9.

**Training Cost.** The number of iterations used for training the LMM is listed in Table 6. The first 80k pre-training iterations were executed without the ADL as well as by evaluating the VGG-perceptual loss over 64-by-64 pixel images. This made each training iteration x6 faster than the following full-resolution and using the ADL. These numbers are lower than the ones reported in (Song et al., 2023), 800k for CIFAR-10 and 2400k for ImageNet 64×64, and in (Kim et al., 2024), 100k for CIFAR-10 and 120k for ImageNet 64×64. We note that these methods rely on having a pre-training DSM as in our case. Training the CFM (Lipman et al., 2023) does not require a pre-existing model, and uses 195k iterations for CIFAR-10 and 628k for ImageNet 64×64. The numbers of iterations quoted here are normalized to a batch size of 512.

Unlike the rest of these methods, the training data of the LMM must be first generated. As noted above, it consists of pairs of the form $x_0, N^*_{\text{Sampler}}(x_0)$ which are sampled from the EDM model $N^*$, in (Karras et al., 2022). The number of training examples we use for each dataset are listed in Table 6. On one hand this sampling process uses fairly high NFEs (35 for CIFAR-10, and 79 for ImageNet 64×64 and AFHQ 64×64), but on the other hand it consists of feed-forward executions with no back-propagation calculations. Moreover, this process can be executed on single GPUs and be trivially parallelized across multiple machines. In terms of wall-clock time this pre-processing did not take long, namely, half a day for CIFAR-10 compared to the 4 days of LMM training, and six days for ImageNet 64×64 compared to 20 days of training, and two days for AFHQ 64×64 compared to 6 training days. We remind that these training sessions were conducted on four GeForce RTX 2080 Ti GPUs.

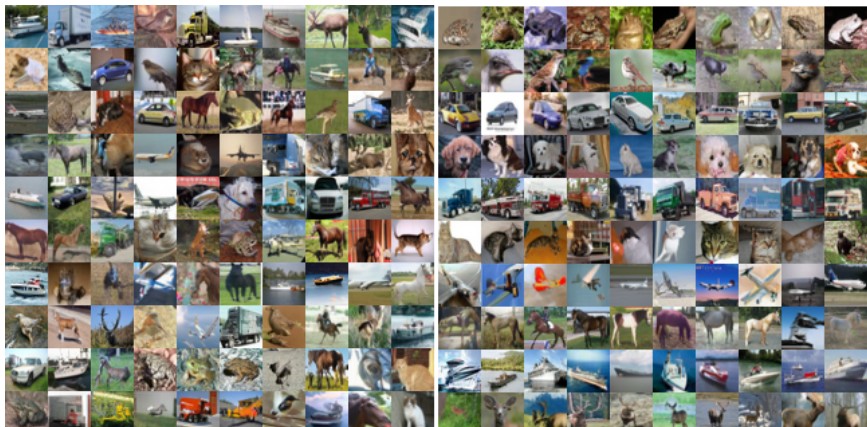

Figure 1: LMM Generated CIFAR-10 Samples. Class unconditional on the left, and conditional on the right. Rows correspond to different classes.

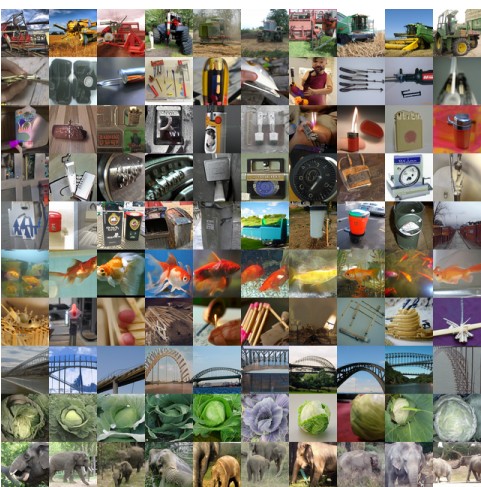

Figure 2: LMM Generated Conditional ImageNet 64×64 Samples. Rows correspond to different classes.

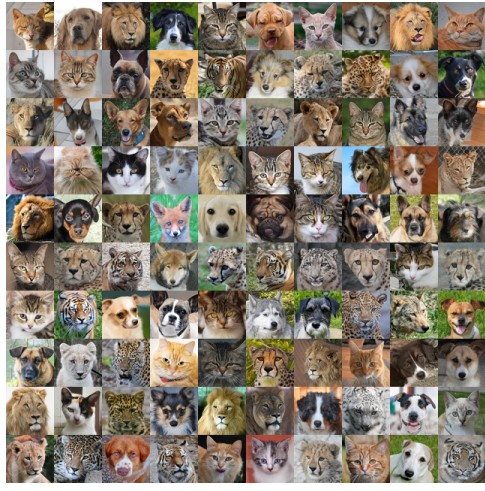

Figure 3: LMM Generated AFHQ 64×64 Samples.

Table 6: Network architectures and hyper-parameters used for different datasets.

| Hyper-Parameter | CIFAR-10 | AFHQ 64×64 | ImageNet 64×64 |
|---|---|---|---|
| Generator architecture | DDPM++ | DDPM++ | ADM |
| Channels | 128 | 128 | 192 |
| Channels multipliers | 2, 2, 2 | 1, 2, 2, 2 | 1, 2, 3, 4 |
| Residual blocks | 4 | 4 | 3 |
| Attention resolutions | 16 | 16 | 32, 16, 8 |
| Attention heads | 1 | 1 | 6, 9, 12 |
| Attention blocks in encoder | 4 | 4 | 9 |
| Attention blocks in decoder | 2 | 2 | 13 |
| Generator optimizer | RAdam | RAdam | RAdam |
| Discriminator optimizer | RAdam | RAdam | RAdam |
| Generator learning rate | 0.0004 | 0.0001 | 0.000008 |
| Discriminator learning rate | 0.002 | 0.002 | 0.002 |
| Generator $\beta_1, \beta_2$ | 0.9, 0.999 | 0.9, 0.999 | 0.9, 0.999 |
| Discriminator $\beta_1, \beta_2$ | 0.5, 0.9 | 0.5, 0.9 | 0.5, 0.9 |
| Batch size | 512 | 512 | 512 |
| EMA | 0.999 | 0.999 | 0.999 |
| Training images | 1M | 2M | 4M |
| Training iterations | 80k+20k w/ADL. | 80k+25k w/ADL | 80k+30k w/ADL |
| $\lambda_{\text{lines}}$ | 0.5 | 0.5 | 0.5 |

| | | CIFAR-10 (conditional) | | | |
|---|---|---|---|---|---|
| NFE | Steps | L2 FID ± std | VGG FID ± std | VGG+ADL FID ± std | VGG+ADL+SOT FID ± std |
| 1 | 0 | $5.125 \pm 0.050$ | $3.124 \pm 0.024$ | $1.672 \pm 0.018$ | $1.575 \pm 0.016$ |
| 2 | 0, 1 | $4.289 \pm 0.032$ | $2.796 \pm 0.020$ | $1.394 \pm 0.010$ | $1.389 \pm 0.011$ |
| 3 | 0, 1, 2 | $4.019 \pm 0.026$ | $2.761 \pm 0.021$ | $1.386 \pm 0.009$ | - |
| 3 | 0, 3, 5 | $3.337 \pm 0.042$ | $2.601 \pm 0.019$ | $1.381 \pm 0.015$ | - |
| 4 | 0, 1, 3, 5 | $3.315 \pm 0.023$ | $2.625 \pm 0.025$ | $1.383 \pm 0.012$ | - |

Table 7: Selected step indices $t$ from the original EDM schedule $\sigma_t$ consisting of 18 steps for this dataset.

| | | ImageNet 64×64 (conditional) | | |
|---|---|---|---|---|
| NFE | Steps | VGG FID ± std | VGG+ADL FID ± std | VGG+ADL+SOT FID ± std |
| 1 | 0 | $6.968 \pm 0.051$ | $1.731 \pm 0.013$ | $1.473 \pm 0.016$ |
| 2 | 0, 1 | $5.472 \pm 0.042$ | $1.318 \pm 0.013$ | $1.167 \pm 0.016$ |
| 3 | 0, 1, 2 | $5.004 \pm 0.057$ | $1.301 \pm 0.012$ | - |
| 3 | 0, 3, 5 | $4.694 \pm 0.047$ | $1.284 \pm 0.016$ | - |

Table 8: Selected step indices $t$ from the original EDM schedule $\sigma_t$ consisting of 40 steps for this dataset.

| | | AFHQ 64×64 | | |
|---|---|---|---|---|
| NFE | Steps | VGG FID ± std | VGG+ADL FID ± std | VGG+ADL+SOT FID ± std |
| 1 | 0 | $5.458 \pm 0.053$ | $2.687 \pm 0.046$ | $2.767 \pm 0.056$ |
| 2 | 0, 1 | $4.254 \pm 0.039$ | $1.545 \pm 0.023$ | $1.776 \pm 0.022$ |
| 3 | 0, 1, 2 | $4.165 \pm 0.045$ | $1.462 \pm 0.016$ | - |
| 3 | 0, 3, 5 | $3.919 \pm 0.035$ | $1.447 \pm 0.023$ | - |

Table 9: Selected step indices $t$ from the original EDM schedule $\sigma_t$ consisting of 40 steps for this dataset.

| EDM | LMM | | | | EDM | LMM | | | |
| --- | --- | --- | --- | --- | --- | --- | --- | --- | --- |
| | wo/ADL | | w/ADL | | | wo/ADL | | w/ADL | |
| NFE 35 | NFE 1 | NFE 2 | NFE 1 | NFE 2 | NFE 79 | NFE 1 | NFE 2 | NFE 1 | NFE 2 |

Table 10: CIFAR-10 (left) and AFHQ 64×64 (right) Samples Comparison.

