# OpenReview forum: "Generative Lines Matching Models"
_ICLR.cc/2025/Conference — ICLR 2025 Conference Withdrawn Submission_

### Official Review · Reviewer_rVCH · 2024-10-27

**Soundness:** 2
**Presentation:** 3
**Contribution:** 2
**Rating:** 5
**Confidence:** 4

**Summary:**

This paper proposes the Line Matching Model (LMM), a probability flow model trained with paired clean and noise data. Given $x_0$ from noise distribution, the authors use a pretrained diffusion model plus deterministic ODE-sampler to propose corresponding clean data $x_1$. The paired $x_0$ and $x_1$ are used to train the LMM using denoising loss. Together with perceptual loss and adversarial Loss, LMM achieves good performance on Cifar-10 (32x32), ImageNet (64x64) and AFHQ (64x64).

**Strengths:**

The paper is well written. The motivation is clear and proposed technique is reasonable.

**Weaknesses:**

1.  The main idea of LMM is to utilize the ability of pretrained diffusion model to build matched data pairs that replace the random selected data pairs used in the original diffusion models. To me, this idea seems to have been already proposed and explored by many previous distillation works like [1,2,3]. The difference is that, works like [1,2] also consider the computational cost of this distillation. Thus, they only run one or two denoising steps instead of running the denoising process till the end to get clean samples. It is not surprising to me that when more computation is allowed, running the denoising towards clean samples can give better teaching signal to the new model.

2. Following 1, although in previous distillation works like [1], distilling diffusion model progressively might cause the error to accumulate, it is more computationally achievable to do so. Directly running the denoising process till clean samples might be achievable on low resolution datasets. It can be unaffordable on high resolution and more complex datasets. And when the data becomes even more complex, more ODE steps will be needed to generate the samples. This further increases the total data preparation time. As mentioned in the Appendix, to save training time, the authors need to pre-generate the samples for LMM. According to Table 6, the size of the generated dataset can be much larger than the size of the original dataset. For example, 1M samples are used for the Cifar-10 dataset while the original dataset only has 50K samples. This means more memory are needed to store the generated dataset.  I'm afraid the high memory cost can be formidable for more complex tasks like text-to-image generation, preventing the proposed method to be scaled up.

3. Though in Table 1, LMM shows strong FID scores comparing with other baselines, I note that this results should also be attributed to the usage of perceptual loss and adversarial loss. According to Table 7 in the appendix, the conditional generation FID scores for Cifar-10 with pure L2 loss are 5.125 (NFE=1) and 4.289 (NFE=2). If we directly brings these numbers to Table1, they do not seem to be very strong. And I would expect worse scores in the unconditional generation cases. Besides, the perceptual loss might be hard to be directly applied to the widely used latent diffusion models. In latent diffusion models, the denoising happens in the latent space, but the perceptual loss is defined in the image space. Decoding noisy latent data to image space to compute the perceptual loss at every training step can be inaccurate and expensive in both time and memory. This might further limited the usage of the proposed method.

[1] Progressive distillation for fast sampling of diffusion models
[2] Consistency models.
[3] Flow straight and fast: Learning to generate and transfer data with rectified flow.

**Questions:**

Please check the weakness.

**Details Of Ethics Concerns:**

No ethics concern

---

### Official Review · Reviewer_mJzy · 2024-10-28

**Soundness:** 2
**Presentation:** 2
**Contribution:** 2
**Rating:** 3
**Confidence:** 3

**Summary:**

This paper proposes a new distillation model that support strong small-step generation quality using the DDIM-based coupling.

**Strengths:**

The paper provides a new perspective on straight trajectory.

**Weaknesses:**

- Let's focus on the setting that $p_0(x_0)$ is the prior distribution. Then, the author's claim is that, with the coupling of $(x_0,x_1)$, where $x_1$ is the solution of PF-ODE, works better than previous algorithms. What does the neural network learns? Does it learn some trajectory? Which trajectory does it learn? How to sample with multi-steps? If it sample like multistep CM, then what's the whole point of having straighter trajectories?

- I personally am not fully convinced of the whole story about the straight flow concept. Basically, what straighter flow promises is that they perform better with small-step samples, but for me, diffusion+distillation seems enough. Given diffusion+distillation, why do we need this straighter flow?

**Questions:**

-

---

### Official Review · Reviewer_JZxz · 2024-11-02

**Soundness:** 3
**Presentation:** 3
**Contribution:** 2
**Rating:** 3
**Confidence:** 3

**Summary:**

This work proposes a diffusion distillation method that aims to establish pairings between samples from the prior and clean data by leveraging ODE solving with a pre-trained model to obtain the clean data. Combined with adversarial training.

**Strengths:**

The presentation is clear and easy to follow. The comparisons of baseline approaches illustrated in Table 1 are both informative and interesting.

I appreciate the analysis in Appendix A.2. Wrapping it up as a theorem would be even clearer to understand the result and claim.

**Weaknesses:**

1. Please be mindful of repeated notations. For instance, $N$ is used both as the maximum timestep and to denote the neural network while ignoring inputs.

2. The proposed method advocates for directly matching the network $ N_{\theta}(\cdot, \sigma)$ prediction with the generated sample from the teacher model $N^*_{\text{sampler}}$ in Eq. (3). However, creating the pairs $ (x_0, N^*_{\text{sampler}}(x_0)) $ may be expensive (offline and online), as it involves solving ODEs. Please also compare the estimated training time.


3. The novelty of the proposed method appears to be limited. Please clarify the fundamental differences between the proposal and [1] as well as CTM. Notably, [1] also suggests matching the network prediction with the sample by solving ODEs until reaching the clean space. Furthermore, the proposal can be viewed as a specific instance of CTM's *soft matching*, where it aligns with what they refer to as *global matching*.

4. [2] also proposed preparing paired data, specifically data-to-noise pairs $(\text{DDIM}^{0\rightarrow T}(x_1), x_1)$ with $ x_1 \sim p_{\text{data}} $, for distillation alongside adversarial training. In this context, what is the fundamental difference between the proposed method and [2]? Additionally, they demonstrated in Fig. 5 that using an imperfect teacher model for noise-to-data collection, combined with adversarial training, results in suboptimal performance. How can we ensure that the proposed method (along with adversarial training) effectively learns the true data distribution?


5. What if the teacher model $N^*_{\text{sampler}}$is not perfect? Including an ablation study based on an imperfect teacher would be beneficial.

6. Given that the training relies on GAN, how can we stabilize its training? Additionally, is the adversarial loss applied from the very beginning of the LMM training? How can we ensure that the model does not suffer from low diversity when using GAN?

7. The discriminator structure of StyleGAN-XL [3] incorporates ImageNet pre-trained feature extractors, which can significantly bias the FID measurements. I recommend comparing all generation quality using EDM-2's proposed Fréchet distances with DINOv2 or training the discriminator for GAN loss from scratch.

Minor comment: The figure in Table 3 appears to be compressed, which may affect the evaluation of the ablation study.


[1] Eric Luhman and Troy Luhman. Knowledge distillation in iterative generative models for improved sampling speed.  2021. URL https://arxiv.org/abs/2101.02388.

[2] Kim, D., Lai, C. H., Liao, W. H., Takida, Y., Murata, N., Uesaka, T., ... & Ermon, S. (2024). PaGoDA: Progressive Growing of a One-Step Generator from a Low-Resolution Diffusion Teacher. arXiv preprint arXiv:2405.14822.

[3] Sauer, A., Schwarz, K., & Geiger, A. (2022, July). Stylegan-xl: Scaling stylegan to large diverse datasets. In ACM SIGGRAPH 2022 conference proceedings (pp. 1-10).

**Questions:**

Please refer to the weakness section.

---

### Official Review · Reviewer_YaF9 · 2024-11-04

**Soundness:** 2
**Presentation:** 2
**Contribution:** 2
**Rating:** 3
**Confidence:** 4

**Summary:**

The paper proposes generative line matching models for few-step image generation. The model is trained on straight lines that connect noise-data pairs produced by ODE sampler, combined with tricks including domain-specific loss, adversarial loss and sample-optimized training. Experiments on CIFAR-10, ImageNet 64×64 and AFHQ 64×64 achieve better 1-step and 2-step FID than previous works.

**Strengths:**

- The pairs produced by diffusion ODE provide clearer signals than the diffusion model prediction, and the perceptual loss/adversarial loss are common techniques targeting the enhancement of FID. It is no surprise that they can bring improvements
- The 1-step and 2-step results on standard image datasets are better than previous works.

**Weaknesses:**

- A large portion of the writing is to explain the flaw of diffusion models in few-step generation and the advantage of straightness. However, these ideas are largely from previous works like Flow Matching, Stochastic Interpolants, Rectified Flow, Consistency Models, and are well-known to the audience who are familiar with diffusion models. The authors should simplify this part and focus on the difference with prior work.
- The authors make claims which seem to convince the readers that diffusion models are incorrect (using words like "singularity" "collapse" and "underestimate"). However, the optimal denoiser is exactly the posterior mean. Diffusion models themselves are well-defined in theory, and it can only be said that "they are not originally designed for few-step generation". The idea of distilling the ODE trajectory is already proposed in consistency models.
- The authors stress the importance of straightness throughout the whole paper, which constitutes the core motivation and idea of the methods proposed. However, this is neither sound nor supported by experiments:
  - Straight path is only a special noise schedule in diffusion models. Different noise schedules are actually equivalent by some transformation [1], and straightness is also not necessary in diffusion distillation, as pointed out by [2].
  - The main improvement of FID stems from perceptual/adversarial loss, while these tricks have nothing to do with the trajectory straightness. In particular, perceptual loss is not preferred as improvements in FIDs can come from accidental leakage of ImageNet features from LPIPS, causing inflated FID scores [3]. Besides, the adversarial loss can perturb the fidelity to the ODE solution.
  - The sample-optimized training restricts the training to only the timesteps at sampling. In 1-step sampling, the intermediate timesteps are never used, so it does not matter whether the constructed path is straight or not. Given the notable improvement in 1-step FID, it can be concluded that it comes from other tricks instead of straightness, the main claim of the paper.

[1] Variational diffusion models
[2] Rectified Diffusion: Straightness Is Not Your Need in Rectified Flow
[3] IMPROVED TECHNIQUES FOR TRAINING CONSISTENCY MODELS

**Questions:**

See weaknesses.

---

### Note · Authors · 2024-11-14

**Comment:**

We thank the reviewers for their time and effort put in evaluating our paper.

We carefully read and considered your comments and came to the conclusion that the way the paper is presented failed to resonate its core contributions, namely,
- the analysis revealing the source of the unwanted basins of attraction in the flow field,
- the complexity analysis of OT-based pairing, and,
- the merit in deriving a flow model with close to straight flows which allows to trade-off quality and NFEs, as opposed to single-step distillation approaches.

In view of that we humbly withdraw our submission for further improvement.

**Withdrawal Confirmation:**

I have read and agree with the venue's withdrawal policy on behalf of myself and my co-authors.